# Potential of TiN/GaN Heterostructures for Hot Carrier Generation and Collection

**DOI:** 10.3390/nano12050837

**Published:** 2022-03-02

**Authors:** Blake S. Simpkins, Sergey I. Maximenko, Olga Baturina

**Affiliations:** Naval Research Laboratory, Washington, DC 22032, USA; sergey.maximenko@nrl.navy.mil (S.I.M.); olga.baturina@nrl.navy.mil (O.B.)

**Keywords:** titanium nitride, gallium nitride, plasmonics, contacts, high-power electronics

## Abstract

Herein, we find that TiN sputter-deposited on GaN displayed the desired optical properties for plasmonic applications. While this is a positive result indicating the possible use of p- or n-type GaN as a collector of plasmonically generated hot carriers, the interfacial properties differed considerably depending on doping conditions. On p-type GaN, a distinct Schottky barrier was formed with a barrier height of ~0.56 eV, which will enable effective separation of photogenerated electrons and holes, a typical approach used to extend their lifetimes. On the other hand, no transport barrier was found for TiN on n-type GaN. While the lack of spontaneous carrier separation in this system will likely reduce unprompted hot carrier collection efficiencies, it enables a bias-dependent response whereby charges of the desired type (e.g., electrons) could be directed into the semiconductor or sequestered in the plasmonic material. The specific application of interest would determine which of these conditions is most desirable.

## 1. Introduction

Refractory metals are typified by high-temperature stability, chemical inertness, and mechanical hardness. Refractory nitrides, which combine a refractory metal with nitrogen, retain the desirable properties of temperature and structural robustness, although they can be susceptible to oxidation. Refractory nitrides have made unique contributions in a number of scientific and technological areas including electronic contacts for high temperature [1,2] or power devices or structures requiring temperature cycling for dopant activation, structural annealing, or driving diffusion processes [3,4]. More recently, they’ve found use in resonant optical applications [5,6] where their thermal stability enables entirely new optically driven applications such as photothermal transducers [7,8,9]. Titanium nitride (TiN), in particular, has attracted attention owing to optical properties that can closely mimic those of elemental gold and a composition that makes it compatible for electronic contacts to semiconducting nitrides.

Important work has been carried out evaluating the interfacial properties in TiN/GaN systems as a function of multicomponent layer composition, annealing conditions, and semiconductor doping. Results have indicated that either pure Ti or TiN can form ohmic contacts to n-GaN, aided by the similarity in their work function and electron affinity. However, the details of specific resistance and interfacial composition for depend on annealing conditions and the heterolayer film structure [10,11,12,13]. On the other hand, TiN does not typically form an ohmic contact to p-GaN, likely due to band alignment issues [1,11,14]. The nature of the interfacial electronic structure, rectifying or ohmic, is critical to charge transport in electronic and optoelectronic devices. Similarly, it has long been appreciated that interfacing a plasmonic resonator with a semiconductor can result in enhanced lifetime and collection of plasmon-derived hot carriers [15]. However, the electronic structure of the metal/semiconductor interface can dictate the efficiency of this charge separation process [16,17]. TiN plays a unique role in this context. Since its optical response can resemble that of gold’s [18], one can implement a variety of TiN plasmonic structures (cubes [19], disks [20], ridges [21], planar films [22]) that exhibit resonant plasmonic response comparable to that of gold, while at the same time, offering the temperature and physical robustness of a refractory material, and a well-controlled interfacial electronic structure with a nitride semiconductor. This combination of traits has made TiN an appealing candidate for photothermal applications or technologies that require large amounts of inexpensive plasmonic material. Furthermore, refractory metallic nitrides comprise a family of materials enabling unique heterostructure design [23,24] for tailored charge generation and transport [25,26] in order to overcome some of the challenges of plasmonic catalysis [27].

In addition to the above-mentioned applications, knowledge of the electronic properties of the TiN/GaN interface can be directly applied to the design of photo-electrochemical systems based on plasmonic nanoparticle/semiconductor interfaces. While interfaces that form a Schottky barrier can perform well, both in the absence and presence of applied bias, applications of photocatalysts that form ohmic contacts are typically limited to photoelectrochemical interfaces only. For instance, TiN nanoparticles supported on semiconducting TiO_2_, which form an ohmic interface, demonstrate activity towards photoelectrochemical water splitting [28] and methanol oxidation, [25] but lack activity towards driving photooxidation of 2-chloroethyl ethyl sulfide molecules at the photocatalyst/air interface (unpublished results). Identifying semiconductors that form a Schottky barrier with TiN nanoparticles can extend their application to photochemical reactions that are not compatible with applied biases, such as an open architecture interfaced with air.

It is with these qualities, challenges, and potential applications in mind that we examine the (1) structural characteristics, (2) optical response applicable to preparing resonant plasmonic structures, and (3) interfacial electronic structure of the TiN/GaN interface for both p- and n-doped GaN without any post-deposition annealing. All of these characteristics play important roles in plasmonically-enhanced photocarrier generation and collection.

## 2. Methods

The samples consisted of ~200 nm of TiN deposited via magnetron sputtering on GaN grown on double-side polished sapphire wafers. The GaN substrates were sonicated in acetone then isopropyl alcohol (5 min each) then dried under N_2_ flow. The TiN films were sputter deposited from a 99.995% Ti target at a substrate temperature of 850 °C with gas flow rates of 2 sccm Ar and 8 sccm N_2_ at a base pressure of 3 × 10^−8^ Torr [29]. The GaN films were commercially acquired from MSE Supplies (n-type Si-doped GaN was 5.69 μm thick with doping of 5.3 × 10^18^ cm^−3^, p-type Mg-doped GaN was 4.6 μm thick with doping of 1.7 × 10^17^ cm^−3^).

The materials were structurally characterized with scanning electron microscopy (SEM) and Raman scattering spectroscopy. Surface structure and roughness were measured for the bare GaN, the deposited TiN, and the etch-revealed GaN surfaces using a Bruker FastScan atomic force microscope. Wet-etching of the TiN was performed by submerging the sample in a solution of 1:2:5 NH_4_OH:H_2_O_2_:H_2_O at 80 °C [30]. Deposited TiN films were characterized optically using an n&k spectrometer in reflection mode after first measuring the GaN/Al_2_O_3_ wafer, then measuring the entire stack with the GaN/Al_2_O_3_ properties fixed to their previously-measured values. These spectroscopic results yielded the frequency-dependent dielectric function of the material, which could then be used to analytically calculate plasmonic response in the small particle dipole approximation. Characterization of the electronic structure of the TiN/GaN interface was accomplished via current-voltage measurements of circular TiN contacts (diameter = 200–400 μm) formed on the GaN surface. Ohmic contacts [31] were formed by electron-beam deposition of Ti/Au (50/100 nm) for n-GaN and Ni/Au (50/100 nm) for p-GaN layers without further annealing and were positioned on the edges of the samples. Measurements were carried out in a dark box probe station with a Keithley model 6487 picoammeter/voltage source system.

## 3. Results and Discussion

### 3.1. Microstructure, Surface Topography, and TiN Etching Analysis

Our discussion of the structural properties of the TiN film begin with SEM imaging and Raman spectroscopy. The SEM micrograph in Figure 1a shows several important structural features of the sputter deposited TiN film. First, we can directly estimate the film thickness from this image to be ~200 nm. Secondly, a columnar structure within the TiN film is evident and likely associated with a growth mode beginning with discrete nucleation followed by expansion and coalescence of these grains into a polycrystalline film. We note that there are no obvious pits or voids (as discussed later and revealed by AFM data).

The structural quality of the TiN film was also evaluated by Raman scattering spectroscopy (Figure 1b). In these spectra (red and blue curves correspond to p- and n-GaN, respectively), signature peaks are visible and correspond to low-frequency acoustic phonon modes (TA and LA located at ~200 and 300 cm^−1^) as well optical phonon modes near 540 cm^−1^ [32]. We also observed combination modes ~410 and 800 cm^−1^. Most important to the work presented here are three points. First, the clear observation of these peaks indicates reasonable crystallinity. Secondly, the positions of the acoustic phonon features at 200 and 300 cm^−1^ and the LO/TO feature at ~540 cm^−1^ are all consistent with low N vacancy concentrations (~0.5%) [33]. Lastly, the relatively strong intensity of the TO/LO peak has been associated with Ti vacancies which can arise due to Ti oxidation [34]. Even though we do not present direct measure of Ti oxidation levels, it is reasonable to expect Ti, a strong oxygen getterer, to exhibit some level of unintended oxidation. We have also presented the Raman scattering response of the bare GaN substrate (dashed curve in Figure 1b) simply to show that no peaks in the TiN spectra can be attributed to the underlying GaN substrate.

The surface structure of a substrate can affect the structural, optical, and electrical properties of subsequently deposited materials. Data acquired by AFM are shown in Figure 2 and reveal the surface structure of as-grown GaN, the TiN films sputtered on those substrates, and the GaN material revealed after wet etching of the TiN. Wet etching was required to produce the patterns used for electrical measurements and can also be used to form top-down plasmonic structures in either single element or arrayed formats. The AFM images in Figure 2a–c are all presented with the same height scaling to enable direct comparison (total height scale is 60 nm) and show considerable variations between the samples. The as-deposited GaN is smoothest, the sputtered TiN film the roughest, and the etch-revealed material exhibited a roughness in between those two extremes. These roughness trends, while not entirely surprising, are important for determining what substrates or deposition conditions are suitable for producing functional TiN overlayers and for determining what size scale of optically resonant structure may be fabricated from such a film. Close examination of the AFM image in Figure 2b shows typical topographic features that are 10–50 nm in diameter. We do not know if these are individual particulates, single crystals, or simply growth mounds, however, their size is comparable to commercially available TiN nanoparticles and would roughly determine the minimum sized feature one could pattern in these films for plasmonic purposes.

The section profiles presented beneath each AFM image are included to give a more quantitative representation of the vertical and horizontal scales of the features on each surface (section locations are indicated with a horizontal dashed line in each AFM image). Note that each of these profiles is scaled uniquely to best reveal each surface’s topography. We will make specific reference to the profile of the TiN film (Figure 2b), which shows considerable roughness (RMS = 13.1 nm which is more than an order of magnitude greater than that of the as-grown GaN), as well as pits. Three of the most prominent pits are identified by arrows in the AFM image and an arrow in the section profile. These data suggest that the pits are 10′s of nanometers deep, although the AFM scanning tip shape can make feature depth analysis inaccurate. This approximate depth is considerably less than the TiN total thickness. Furthermore, the experimental observation of a Schottky interface between the TiN film and p-GaN (discussed later) indicates there are no direct current leakage paths accessed during the IV measurements. An estimation gives the density of these pits as ~3 × 10^8^ cm^−2^, however it’s unclear if they are all of similar depths or if some penetrate deeper.

Lastly, we comment on the wet etching process used to selectively pattern the TiN film. Figure 2c shows a topographic image, section profile, and RMS value for such an etch-revealed surface. While this surface is notably smoother than the sputtered TiN film, we did not see a complete return to the condition of the pristine as-grown GaN surface. The image reveals features which may be residual TiN which is not entirely removed. This supposition is supported by the section profile which shows an apparent smooth baseline underneath topographic features (baseline highlighted by dotted red line in Figure 2c). The wet etch used (1:2:5 NH_4_OH:H_2_O_2_:H_2_O at 80 °C) removed TiN ~linearly (Figure 2d) at an etch rate of 0.87 nm/s, and indicated a total TiN film thickness of ~160 nm. This is ~25% smaller than the thickness measured with cross-sectional SEM (see Figure 1a). This could be due to heterogeneity in film thickness or a slight off normal cross-sectional orientation when imaging Figure 1a, which would increase the apparent film thickness measured via SEM imaging.

### 3.2. Dielectric Characterization and Optical Implications

Paramount to plasmonic or other optical use of these systems is their dielectric response. It is known that the complex refractive index and electrical conductivity of TiN is closely tied to its deposition and processing conditions and the resulting structural and defect characteristics [35,36]. Since the materials under investigation here were sputtered onto GaN, with its own surface structure as discussed above, it is essential to measure the optical response of these films to verify their utility for plasmonic applications. Therefore, we carried out direct measurements of the complex dielectric function of the TiN and used these measured values to predict the expected plasmonic resonances for ellipsoids with an increasing aspect ratio. First, notice the measured values of the real and imaginary components, *ε*_1_ and *ε*_2_, of the dielectric function in Figure 3a. These plots immediately tell us where the scattering resonance will reside for a sphere in air (i.e., the frequency where *ε*_1_ = −2) and allow for the extraction of the plasma frequency and damping constant according to Drude modeling of the material [37]. The real and imaginary parts of the complex dielectric function (Equation (1)) can be separated to obtain two linear expressions, Equations (2) and (3).
(1)$$\varepsilon =1-\frac{\omega_{p}{}^{2}}{\omega^{2}+i\omega \gamma_{d}}$$
(2)$$\omega \varepsilon_{2}=\gamma_{d}\left(1-\varepsilon_{1}\right)$$
(3)$$\omega^{2}\left[\varepsilon_{2}{}^{2}+{\left(1-\varepsilon_{1}\right)}^{2}\right]=\omega_{p}{}^{2}\left(1-\varepsilon_{1}\right)$$

In these expressions, *ε* is the full complex dielectric function, *ε* = *ε*_1_ + *i**ε*_2_, *ω**_p_* the plasma frequency, *γ**_d_* the damping constant, and *ω* the frequency of interest. One can see that the plasma frequency and damping constant can be extracted from the slopes of Equations (2) and (3). Such plots are presented in Figure 3b and yielded a plasma frequency, *ω**_p_* = 1.23 × 10^15^ s^−1^ (i.e., wavelength of 243 nm) and a damping constant of *γ**_d_* = 204 THz. These characteristics are key for using metals for optical applications since the plasma frequency represents an upper bound on the frequency at which the material behaves as an ideal metal. Much of the motivation for exploring the optical response of TiN has grown from its optical similarities to Au. For comparison, the plasma frequency extracted above for TiN is ~40% less than for Au and has damping constant about an order of magnitude larger [38]. We add the caveat that the data in Figure 3b are not ideally linear and choice of fit ranges can impact the extracted slope.

Nevertheless, with these parameters in hand, one may model an optical response of interest. Since our efforts have been focused on utilizing hot carriers generated from plasmon decay, [39,40,41] we calculate the absorption for ellipsoids of varying aspect ratio for both the TiN under investigation and Au for comparison. The results, Figure 3c, show that (1) despite increased damping relative to Au, clear optical resonances are predicted to arise in the TiN, (2) these resonances are broader and located at lower energies than Au particles of the same geometry, and, as expected, (3) these resonances tune to the red with increased aspect ratio due to reduced restoring forces for longer rods [42]. These results are consistent with previous examinations of plasmonic resonances in TiN and suggest that TiN sputtered on GaN possesses optical properties sufficient for plasmonic applications.

### 3.3. Schottky Barrier Measurements and Discussion

Current-voltage characteristics of the structures were measured utilizing a dark box probe station and Keithley model 6487 picoammeter/voltage source system at room temperature. Figure 4 shows the typical current density-voltage response of the structures formed on n- and p-type GaN. I–V characteristics typically observed for contacts with ohmic behavior were found for the TiN contacts patterned on n-type GaN. Field emission is the preferred transport mode there, where carriers tunnel through the potential barrier as schematically shown in the inset to Figure 4a. This mechanism occurs when the interfacial depletion layer is sufficiently narrow, a consequence of high doping concentration in the semiconductor, which is the case for this n-type sample with doping of 5.3 × 10^18^ cm^−3^. Contacts formed on p-type GaN showed distinctive nonlinear behavior associated with the presence of a Schottky barrier at the TiN/GaN interface. Thermionic emission modeling [43] is most commonly applied to describe current-voltage characteristics of a Schottky barrier according to
(4)$$I=AA^{*}T^{2}\exp\left(-\frac{q\varphi }{kT}\right)\left(\exp\left(\frac{qV}{nkT}\right)-1\right)=I_{S}\exp\left(\exp\left(qV/nkT\right)-1\right)$$
where *A* is the device area, *A** the Richardson constant, *q* the electron charge, *k* the Boltzmann constant, *T* the temperature, *n* the device ideality factor, and *V* is the applied voltage. The value of the Schottky barrier height (*φ*) can be found by first extrapolating of the log(*I*) versus *V* curve to *V* = 0 to find *I_s_*. Then, the barrier height is described according to the expression below [43].
(5)$$\varphi =\frac{kT}{q}\ln\left(\frac{AA^{*}T^{2}}{I_{s}}\right)$$

Using the experimentally reported Richardson constant of 109 A/cm^2^K^2^ for p-GaN [44], the barrier height was evaluated to be 0.56 eV (see Figure 4b inset plot for schematic of this process). It should be noted that the forward I–V curves from the measured devices had a double humped response, an indication of contact inhomogeneity (i.e., areas on the contact with differing barrier heights).

The results presented above correlate reasonably well with results reported in the literature for TiN on GaN. Ohmic interfaces have been reported for TiN on n-type GaN, however, subsequent annealing was required [13,45], while as-deposited contacts exhibited barrier heights of ~0.5–0.6 eV [46]. The literature also notes that the formation of a TiN interfacial layer, formed by the reaction of Ti and GaN during contact annealing, is necessary for the creation of a high-quality Ti ohmic contacts to n-GaN [47]. Reported data for Schottky contacts to p-type GaN are somewhat scattered, particularly for TiN contacts. What is known, however, is that it is difficult to form high barrier contacts to p-GaN [48] and that the surface and material quality is crucial for the formation of a high-quality Schottky contact [49,50].

## 4. Conclusions

We have evaluated a number of relevant properties of TiN sputtered on p- and n-type GaN. This system is of interest for high-power electronics, photochemical, and optoelectronic applications. Most important for plasmon-generated carrier collection are the properties of surface structure, frequency-dependent dielectric function, and the electronic structure of the TiN/GaN interface. We found significant roughness in the sputtered films, which were characterized by a columnar microstructure, particulates sized 10–50 nm and pits (10’s of nm deep at a density of ~3 × 10^8^ cm^−2^). Raman spectroscopy results suggested low N vacancy levels but some degree of Ti vacancies, likely due to Ti oxidation. Despite this rough morphology and potential inclusion of oxidized Ti, the measured dielectric response indicated this material should support plasmonic resonances located near 700 nm and could be tuned to the red via aspect ratio, dielectric environment, or particle-particle interactions. Lastly, we verified that the sputtered TiN formed an ohmic contact to n-GaN, without the need for annealing, and a rectifying contact with p-GaN with a barrier of 0.56 eV. These results are useful for the design of devices which rely on plasmonically generated hot carriers. In many applications (PV, photodetection), it is desirable to separate electron-hole pairs to enhance collection efficiencies, for which a Schottky contact is preferred. On the other hand, for catalytic applications, where the reaction may preferentially occur on the metal or the semiconductor, the ability to direct carriers across an ohmic interface via bias would be most useful.

## Figures and Tables

**Figure 1 nanomaterials-12-00837-f001:**
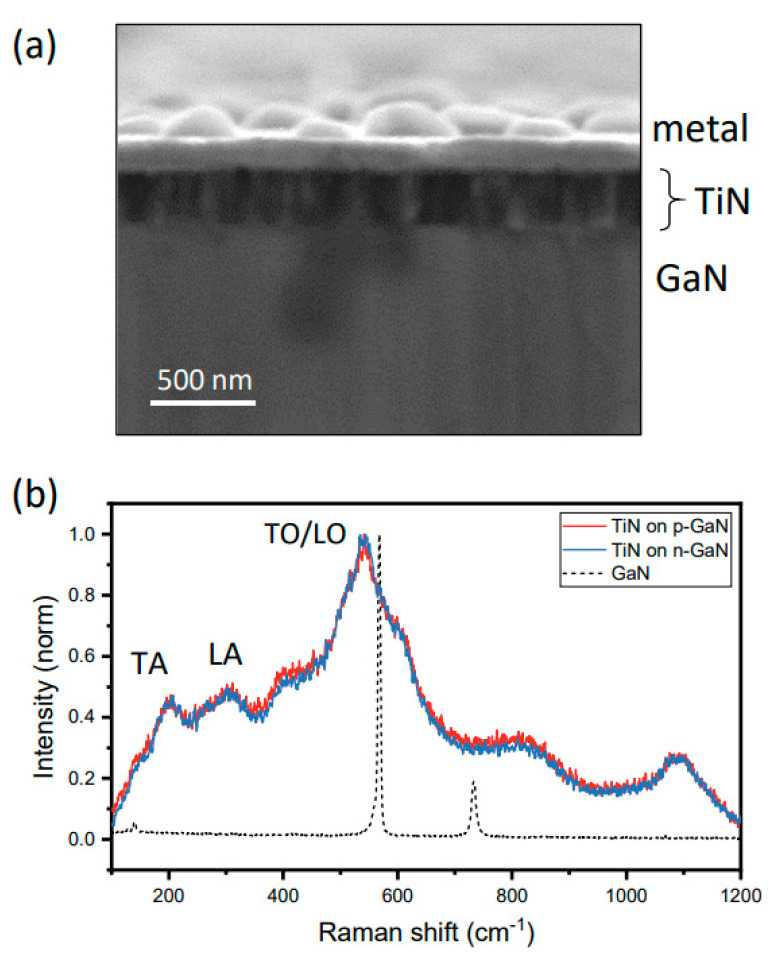
Structural characterization. Cross-sectional SEM shown in (**a**) reveals columnar microstructure of TiN film and thickness of ~200 nm. Raman scattering spectra (**b**) of TiN grown on p- and n-GaN (red and blue) show virtually identical features attributed to dominant optical phonons ~540 cm^−1^ and acoustic phonons ~300 cm^−1^ and below. The response of the GaN substrate is shown as a dashed curve.

**Figure 2 nanomaterials-12-00837-f002:**
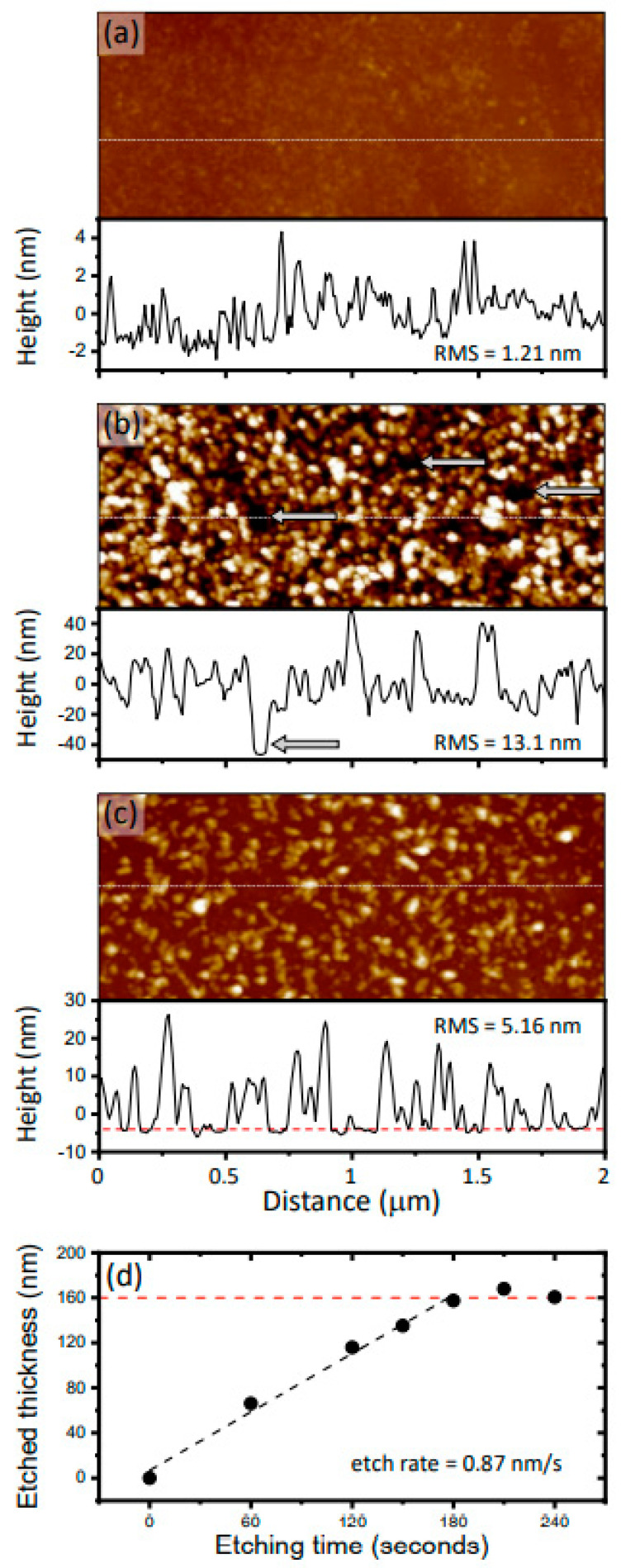
Topographic images and representative section profiles for (**a**) as-grown p-GaN, (**b**) sputter-deposited TiN, and (**c**) etch-revealed GaN. All images shown with total height scale of 60 nm and total horizontal length scale of 2 μm. The RMS roughness values are given in each panel and observed pits in (**b**) are indicated by grey arrows. Location of section profiles indicated by horizontal dashed lines. The etch rate is presented in (**d**) where etch depth vs. etch time is plotted. These data reveal an etch rate of 0.87 nm/s and total TiN film thickness of ~160 nm.

**Figure 3 nanomaterials-12-00837-f003:**
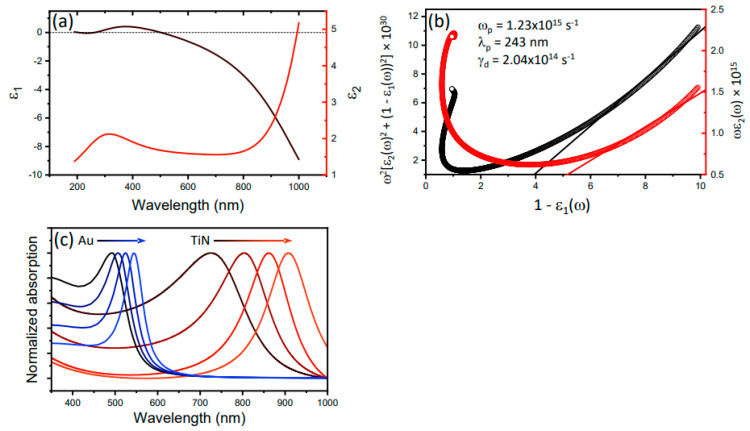
The measured real and imaginary components of the dielectric function are plotted in (**a**). The data is replotting in (**b**) according to Equations (2) and (3) allowing for the extraction of the plasma frequency and damping constant (*ω_p_* = 1.23 × 10^15^ s^−1^ and *γ_d_* = 204 THz). Using the small particle dipole approximation, we calculate the expected plasmonic resonance for ellipsoids with aspect ratios varying from 1 to 2.5 for TiN (black to red) and compare to Au (black to blue) in (**c**).

**Figure 4 nanomaterials-12-00837-f004:**
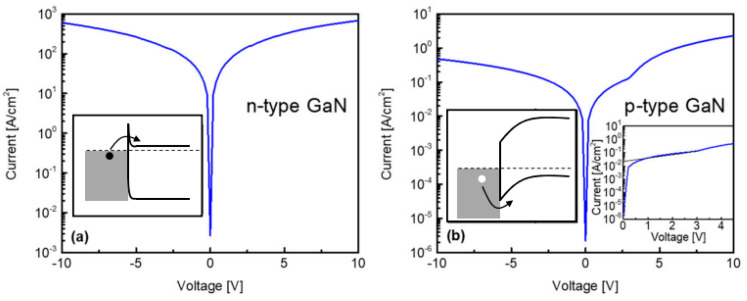
Current-voltage characteristics of TiN contacts (400 mm dia.) fabricated on (**a**) n-type and (**b**) p-type GaN films. A symmetric ohmic-type response was found for TiN/n-GaN in (**a**). Rectifying response was found for TiN on p-type GaN in (**b**) with the inset graph showing the positive portion of the plot fitted to extract a barrier height of 0.56 eV. Each plot includes a schematic illustrating qualitative band structures for each system.

## Data Availability

The data presented in this study are available on request from the corresponding author.

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
