# Peer review of "Potential of TiN/GaN Heterostructures for Hot Carrier Generation and Collection"

_nanomaterials, 2022, doi:10.3390/nano12050837_

Round 1

Reviewer 1 Report

The authors have achieved an interesting result for TiN films, deposited via magnetron sputtering, on top of n/p-GaN thin wafer,. They find that the TiN formed a good quality and performance ohmic contact to n-GaN, without the need for annealing, as has been reported was necessary by similar structures, previously in the literature.

They also find that TiN forms a rectifying contact with p-GaN with a barrier that they model is of 0.56 eV.

As the authors discuss, these results are useful for the design of devices which rely on plasmonically generated hot carriers, and hence it is a result of interest to scientists working in the subject of nanophotonic devices.

In regards to the work I have some observations on several limitations to the present report:

Being the essential merit of the manuscript, the ability of the deposited TiN film on n-GaN to form good quality ohmic contacts, the paper is devoid of any description on the sputtering growth, i.e. growth conditions employed, for instance as well as others necessary to judge on the merits of their result. Also, I think that the materials or device scientist reading the article may find that some additional structural characterization (for instance SEM and Raman may help) are necessary in order to evidence or perceive why their material provided superior performance, in contrast to other reports on similar structures in the literature, which require annealing to provide good quality ohmic contacts.

As the authors discuss correctly that this structures with the properties they have obtained, may be attractive to use them for designing devices operating on plasmonically generated carriers, I found to my surprise that no plasmon absorption spectra were presented in the manuscript, only some theoretical model on how this absorption may result. The deposited samples are relatively thin, 170 nm, but it is deposited on top of GaN thin wafers, 4.6-6 micrometers thick, dimensions that may be enough to experimentally extract the corresponding TiN absorption spectrum in the plasma resonance region, and compare with their theoretical results.

Please comment on this observation.

Reviewer 2 Report

The authors presented their investigation on the optical properties of TiN/GaN for plasmonic applications. The paper is well written and scientifically sound. However some very minor edits should be taken into consideration. Specifically, the figure captions should be on a text form and not part of the image. For Figures 1a-c, the location of the section profile should be indicated in the AFM image. Hence,  I recommend the publication of the paper after minor revisions.

Reviewer 3 Report

The article presents the results of a study of the optical properties of titanium nitride deposited on gallium nitride, which have the potential to use these structures in plasmonics. The authors have established a number of dependencies that are of great scientific interest, in view of their novelty. This research topic is one of the most relevant today, and the selected research objects have a high potential for practical application. However, after reading the reviewer, a number of questions arose that the authors should answer before the work can be published.

1 In the introduction, authors should draw attention to similar studies and provide a brief overview of the current state of research in this area.
2. How exactly the film thickness indicators were determined, the authors should provide a more detailed description. The specified speed value also requires explanation.
3. The authors should explain what is the reason for the shift in the absorption maximum shown in Figure 2.
4. It is also necessary to explain the presence of a maximum of the dielectric function at a wavelength of 250 nm
5. The results of morphological studies require significant improvement.
6. The authors should clarify whether the structural parameters of the synthesized films have been determined.

Round 2

Reviewer 1 Report

The authors have answered in a satisfactory way to all my observations, both in the response letter, and in the revised manuscript. I have no further observations or objections, hence, I'm of the opinion that the paper is suitable to be published in your prestigeous Journal. 

The article may require a moderate English lenguage editting. 

Author Response

We thanks the reviewer for their time and comments.  We have re-edited the manuscript for grammar and clarity.

Reviewer 3 Report

The authors answered all the questions, the article can be accepted for publication.

Author Response

(The authors gave the same response as above.)
